# Dual-stream Feature Augmentation for Domain Generalization

## ABSTRACT

Domain generalization (DG) task aims to learn a robust model from source domains that could handle the out-of-distribution (OOD) issue. In order to improve the generalization ability of the model in unseen domains, increasing the diversity of training samples is an effective solution. However, existing augmentation approaches always have some limitations. On the one hand, the augmentation manner in most DG methods is not enough as the model may not see the perturbed features in approximate the worst case due to the randomness, thus the transferability in features could not be fully explored. On the other hand, the causality in discriminative features is not involved in these methods, which is harm for the generalization of model due to the spurious correlations. To address these issues, we propose a **D**ual-stream **F**eature **A**ugmentation (DFA) method by constructing some hard features from two perspectives. Firstly, to improve the transferability, we construct some targeted features with domain related augmentation manner. Through the guidance of uncertainty, some hard cross-domain fictitious features are generated to simulate domain shift. Secondly, to take the causality into consideration, the spurious correlated non-causal information is disentangled by an adversarial mask, then the more discriminative features can be extracted through these hard causal related information. Different from previous fixed synthesizing strategy, the two augmentations are integrated into a unified learnable model with disentangled feature strategy. Based on these hard features, contrastive learning is employed to keep the semantics consistent and improve the robustness of the model. Extensive experiments on several datasets demonstrated that our approach could achieve state-of-the-art performance for domain generalization.

## CCS CONCEPTS

• **Computing methodologies** → **Transfer learning**; **Object recognition**.

## KEYWORDS

Domain Generalization, Feature Augmentation, Feature Disentanglement

## 1 INTRODUCTION

Deep neural networks [15, 20, 39] have seen widespread integration into various fields, showcasing significant potential for diverse applications. While deep learning models are effective, real-world

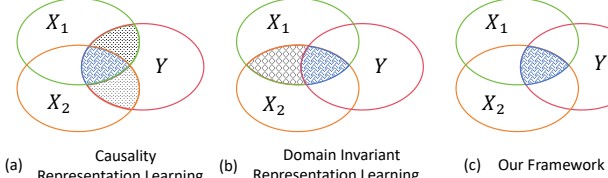

(a) Causality Representation Learning  (b) Domain Invariant Representation Learning  (c) Our Framework

**Figure 1:** $X_1$ **and** $X_2$ **represent two different domains, and** $Y$ **represents the label space shared by both source domains. (a) The dashed areas represent the mathematical statistical relationship between each domain and labels. Obviously, the areas not only include the shared part, but also contains the specific parts. (b) The dashed area represents the domain invariant information across multiple source domains. However, the spurious correlation information still exists in it. (c) Our motivation is to learn domain invariant features that have causal relationships with the labels.**

scenarios often pose challenges such as non-stationary and unknown distributions in testing data. To address distribution shifts between training and testing data, the domain generalization task has emerged. This task aims to make the model robust and generalized across multiple source domains, enabling its application in unknown target domains. In order to obtain the generalizable and accurate features in DG task, the criteria of transferability and discriminability are both important. As shown in Fig 1(a), due to the existence of domain shift, the mathematical statistical relationship between features and labels are different in different domains. Even the model has high discriminability in source domains, it can not work well in target domains due to domain shift. However, only concerned about the transferability is not enough. Shown in Fig 1(b), the dashed area refers to the domain-invariant information learned from multiple domains. Although leveraging domain adversarial learning can get good transferability of features, the model only focuses on the domain shared information, which maybe harm the final downstream tasks. *e.g.*, some non-causal information that has spurious correlations with labels can not be distinguished. Then the model could not generalize well to unseen domains. Based on this, in order to relieve the above issue, we want to achieve the goal shown in Fig 1(c), which could not only eliminate the spurious correlated non-causal information, but also exploit the domain invariant features with sufficient causality.

Data augmentation has been demonstrated effective in DG task recently. Generally speaking, these augmentation methods keep the semantics consistent and modify the style of samples to enhance the diversity. This strategy goes against domain shift and makes the model pay more attention to features that are invariant to domain transfer. If the model could fully focus on capturing the statistical dependency between the semantic information and the corresponding labels, it could eliminate bias toward a particular domain distribution. According to the Empirical Risk Minimization (ERM) principle, to improve the generalization capability of

a model, an effective way is to optimize the worst-domain risk over the set of possible domains. However, despite the performance could be promoted by generating samples that are created to approximate the worst case across the entire family of domains, it is hard to generate "fictitious" samples in the input space without losing semantic discriminative information. Moreover, previous methods always adopt the two-stage data perturbation training procedure, and the perturbed samples can not achieve the self-adaptation with the different samples. Fourier transform [46] is a well-known data augmentation manner and obtains competitive results. Usually, in order to avoid the semantic changes, domain transfer is achieved by adding random noise to the Fourier spectrum amplitude components of the sample, then the new data augmentation can be generated. However, this random way may induce unpredictable alterations in the image style. Minor disturbances might have no significant impact on the domain style, rendering the style transformation ineffective. Conversely, substantial perturbations could distort the image style, potentially affecting its semantics and introducing label noise.

Based on this, we aim to achieve the data augmentation by hard perturbing features without changing the semantics. In order to fully explore the generalization boundaries and avoid the semantic level collapse, instead of samples, the data augmentation in our method is performed on the feature level. We propose to perturb the hard features based on a feature disentanglement framework, as shown in Fig 2. On the one hand, to obtain the transferability, we aim to construct the consistent semantic augmented features with another domain information. As illustrated in Fig 2 left, the domain-specific information with the most abundant style attributes is selected to construct domain related hard features. In information theory, the entropy is an uncertainty measure which can be leveraged to quantify the domain style. However, only rely on reducing the domain difference to improve the generalization is not enough, although the feature disentanglement could guarantee the semantics not be changed, it does not involve the spurious correlation and non-causal information in the features. In DG task, causality is an important factor for the discriminability. On the other hand, to improve the reliablity of the statistical dependence, the spurious non-causal correlations should be eliminated and the invariant causal correlations should be mined. Based on the semantic features of disentanglement, we propose to construct the causal related hard features. As shown is Fig 2 right, the non-causal information with similar labels exhibits spurious correlation with semantics, which could be used to construct causal related hard features. These features consist of the domain-invariant semantics and the spurious correlated non-causal information from another class. The causal and non-causal related information in our method can be separated by an adversarial mask. With the help of the two stream hard features, our model could fully explore the causal factor based on the domain invariant features, thereby the transferability and discriminability of features can be fully preserved.

In this paper, we integrate the two feature augmentation manner into one unified framework and propose a dual stream feature augmentation based on the feature disentanglement framework. In the framework, domain-invariant causal features are obtained through the feature disentanglement strategy with the help of domain related and causal related hard features. To keep the semantics

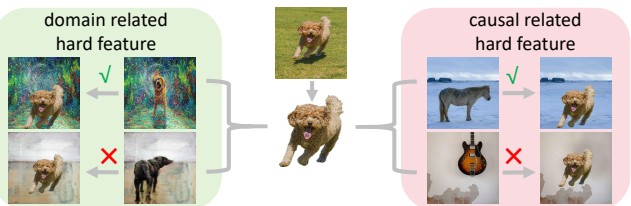

**Figure 2: Diagram of our feature augmentation. For domain related augmentation, the domain-specific information with the most abundant style attributes is selected to construct hard features. For causal related augmentation, the most correlated non-causal information within the most similar class is selected to construct hard features.**

consistent, contrastive learning is leveraged to dual-stream augmented hard features respectively. Noteworthily, the perturbed features are created during the training process, therefore the online features can adapt with the different inputs. The contributions of our work are as follows:

- We point out the disadvantages of present data augmentation methods, and propose a domain related hard feature perturbation strategy with semantic consistency based on an end-to-end stable disentanglement framework, thereby improving the transferability of features.
- To fully explore the discriminability in generalized features, the causality is considered in our method. Causal related hard features are created with the disentanglement framework, thus eliminating the underlying non-causal information hidden in features.
- We conduct extensive experiments on several public benchmarks, which clearly demonstrate the effectiveness of our approach.

## 2 RELATED WORK

**Domain Generalization.** The goal of DG task is to learn the generalizable representations from source domains to ensure stable performance to unknown target domain. Existing methods can be roughly divided into domain-invariant or causal-related feature learning [3, 26, 28, 29], data augmentation [10, 18, 44, 46] and other learning strategies, such as meta-learning [3, 7, 49] and contrastive learning [8, 17, 47]. Domain-invariant representation learning has become an important method in DA and DG since [13] was proposed. This method facilitates the model in learning domain invariant features through min-max adversarial training between the semantic feature extractor and domain discriminator. [3] also employed a dual path strategy, integrating domain-invariant and domain-specific encoders, similar to our approach. However, they trained two domain classifier for two encoder respectively, which still constitutes an adversarial training process. In recent years, there has been increasing interest in investigating domain generalization from the causal perspective. [28, 42, 45] derived causal information that truly determine the category label from the statistical relationship between the sample and the label. [28] analyzed the three fundamental properties that causal factors should satisfy,

thereby achieving the objective by ensuring the learned representations comply with these three properties. However, this may lead the model to learn some domain-specific information from the source domains. An adversarial mask is employed to disentangle spurious correlated non-causal information as in [28]. However, in our work, the mask is applied to domain-invariant features to avoid negative impacts. [17] employed a domain-aware contrastive learning that aims to minimize the distance between stylized and original feature representations. [47] proposed an proxy-based contrastive learning approach. This method used proxies as the representatives of sub-datasets and managed the distance between features and proxies, thereby enhancing the robustness against noise samples or outliers. [8] generated domain-invariant paradigms for each instance and then conducted contrast learning between the features of image instances and their paradigms. In our work, we apply supervised contrastive learning strategy to dual stream augmented features and domain-invariant features. This enables the model to eliminate potential stylistic information and non-causal information inherent in the domain-invariant features, thereby enhancing the model's generalization ability.

**Data Augmentation.** Data augmentation techniques for Domain Generalization (DG) can be broadly categorized into image generation [53, 55], image transformation [41, 46], and feature augmentation [48, 56]. However the offline two-stage image generation training procedure is complex, as both training a generative-based model and inferring it to obtain perturbed samples present significant challenges. [46] perturbed the style of a sample through linear interpolation between the Fourier spectrum amplitude components of the sample. However, it randomly selected the exchange sample and ratio. [48] employed Wavelet Transforms to decompose the features into high and low frequencies. [27, 41] achieved feature style transformation by executing a series of processes on the low-frequency component of features. The statistical properties [33] of the feature maps can represent stylistic information as they capture visual properties, [18, 44, 52, 56] achieved style transformation by perturbing statistics of features. However, these methods directly interfere with feature statistics and often fail to maintain semantic consistency. Very recently, [22] is proposed to explicitly enforce semantic consistency preserving class-discriminative information. It generated learnable scaling and shifting parameters for features to enhance domain transfer from the original ones and this idea is very similar with ours. However, it is essentially still a random augmentation method, while our method aims to generate targeted features. In our method, we construct hard augmented features through DFA, enhancing the generalization capacity of the model. Not only domain style transformation but also causal related information augmentation is implemented in our work.

## 3 METHOD

The source domains $\mathcal{D}_s$ and target domain $\mathcal{D}_t$ share the same label space in DG task. Each source domain consists of $\mathcal{D}_s = \{(x_i, y_i)\}_{i=1}^{N}$, in which $x$ represents sample and $d$ represents label. In our method, we use dual path feature disentangle module to obtain domain-invariant features and domain-specific features. Then the adversarial mask module is introduced, the potential causal information is mined through it to disentangle spurious correlated

non-causal information among domain-invariant features. Finally, we present the dual-stream feature augmentation.

### 3.1 Dual path feature disentangle module

Domain-invariant features refer to the shared semantic characteristics across multiple domains, which remain consistent despite domain shifts. Features that cannot be distinguished by the domain classifier are considered effective domain-invariant features. Ensuring that the domain classifier can accurately identify domain features is crucial. Most methods are updating both the domain-invariant encoder $F_I$ and the domain classifier $C_d$ together. However, training domain classifier with domain-invariant features which do not contain domain-specific information could not guarantee its effectiveness. Therefore, we propose a dual path feature disentangle module, which ensures the accuracy of the domain classifier by leveraging an extra domain-specific encoder. The proposed method consists of a domain-invariant encoder $F_I$, a domain-specific encoder $F_S$ and a domain classifier $C_d$. To achieve an optimal domain classifier, we conduct the training of the domain-specific encoder and the domain classifier as a $k$ classification task, as defined by Eq 1, where $d_i$ represents domain label and $k$ represents the number of source domains.

$$\mathcal{L}_{dc}^{spe} = \ell \left( C_d \left( F_S \left( x_i \right) \right), d_i \right) \tag{1}$$

Regarding the domain-invariant features $f_I$, they are passed through the domain classifier $C_d$ to obtain the domain classification probability $P_{f_I}$. The domain-invariant encoder $F_I$ is then updated by Eq 2. Noteworthily, to force the features not containing domain specific information, instead of the cross entropy loss, we use the mean square error (MSE) loss to make the classification probabilities as smooth as possible. The reason is that the domain classifier is supposed to be impossible to distinguish domain-invariant features.

$$\mathcal{L}_{dc}^{inv} = (P_{f_I} - \frac{1}{k})^2 \tag{2}$$

### 3.2 Adversarial Mask Module

To ensure that the features are causally sufficient and contain more potential causal information, the adversarial mask module [28] is employed to achieve the goal. We aim to categorize the feature dimensions into superior dimensions, which are related to causal information, and inferior dimensions, which lack sufficient causal information and exhibit spurious correlations with the labels. Obviously, the superior dimensions of features have stronger relevance to the semantics than the inferior. Through the adversarial training between two classifiers, the two kinds of features can be separated. Specifically, a neural network $M$ is built, by using derivable GumbelSoftmax to sample the mask $M(x)$. Through multiplying the domain-invariant features with the resulting masks $M_{sup} = M(f_I)$ and $M_{inf} = 1 - M(f_I)$, we can obtain superior and inferior features, respectively, and then feed them into two different classifiers $C_1$ and $C_2$. The optimization process between encoder, classifier and mask is an adversarial learning process. On the one hand, two classifiers and encoder are optimized by cross-entropy loss, so that they can mine more semantic information, as shown in Eq 3. On the other hand, the mask is optimized through adversarial training by maximizing the classification loss of the inferior dimensions, as shown in Eq 4, to better distinguish superior and inferior dimensions.

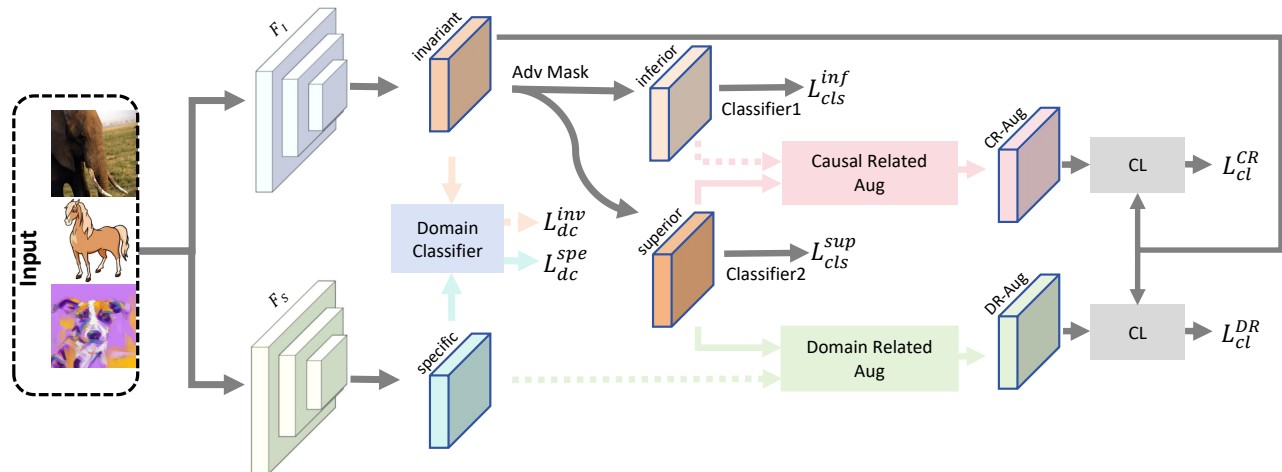

**Figure 3: The framework of DFA. We first generate domain-invariant features and domain-specific features by dual path feature disentangle module, and employ adversarial mask module to disentangle spurious correlated non-causal information from domain-invariant features. We combine superior features with domain-specific information and non-causal inferior information by special strategy respectively to achieve dual-stream feature augmentation. At last, contrastive learning is adopted to the augmented features and domain-invariant features. The dashed lines represent that the gradient is detached.**

$$\mathcal{L}_{cls}^{sup} = \ell(C_1(F_I(x_i) * M_{sup}), y_i)$$
$$\mathcal{L}_{cls}^{inf} = \ell(C_2(F_I(x_i) * M_{inf}), y_i) \tag{3}$$

The overall loss of the adversarial mask module is depicted in Eq 4 and Eq 5.

$$\mathcal{L}_{mask} = \mathcal{L}_{cls}^{sup} - \mathcal{L}_{cls}^{inf} \tag{4}$$

$$\mathcal{L}_{cls} = \mathcal{L}_{cls}^{inf} + \mathcal{L}_{cls}^{sup} \tag{5}$$

### 3.3 Dual-stream feature augmentation

In this section, we will introduce the two types of feature augmentation methods respectively. Assuming that each domain contributes $n$ samples to a batch and there are $k$ source domains in total, the batch size is calculated as $B = n \times k$.

**3.3.1 *Domain related feature augmentation*.** Dual path feature disentangle module enables the model concentrate on domain-invariant information, partial mitigating performance degradation induced by domain shift. However, only rely on feature disentanglement does not ensure that domain-invariant features are completely separated from domain-specific information. Therefore, we propose a domain related feature augmentation to generate fictitious data. According to the Empirical Risk Minimization (ERM) principle, the model could improve the generalization capability by optimizing the worst-domain risk with the perturbed cross-domain features.

To construct domain related hard features, the superior dimensions of domain-invariant features and the most distinct style from the domain-specific features in other domains should be selected. To achieve this goal, information entropy is leveraged as the criterion for evaluating style features, as illustrated in Eq 6. Lower information entropy indicates a more distinct style information. To ensure augmentation diversity, we select one sample from each domain sequentially, using $k$ samples as a group to implement the

aforementioned augmentation. And there are $n$ groups within a single iteration. The chosen domain-specific features and disentangled superior dimensions features are merged via the concatenation operation. Then subsequently the feature dimension is reduced through a fully connected layer, as shown in Eq 7. The $[\cdot, \cdot]$ is a concatenation operation and FC is a fully connected layer. $f'_S$ represents the domain-specific features that belong to other domains in a group.

$$IE(X) = -P(x)log(P(x)) \tag{6}$$

$$f_{DR-Aug} = FC([f_{sup}, min(IE(f'_S))]) \tag{7}$$

Intuitively, the semantic information of domain related hard features aligns with domain-invariant features, and domain-invariant features inevitably retain some aspects of their original domain-specific information. To eliminate potential domain-specific information in domain-invariant features, the supervised contrastive learning is applied to domain-invariant features and domain related hard features, as shown in Eq 8. By drawing the positive samples close and the negative samples separated in the feature space, the transferability of the domain-invariant features can be enhanced.

$$\mathcal{L}_{cl}^{DR} = \ell_{cl}(f_I, f_{DR-Aug}) \tag{8}$$

**3.3.2 *Causal related feature augmentation*.** Due to the limitations of the dataset and the insufficient diversity of samples, the model would inevitably learn some spurious correlated non-causal information when capturing the statistical relationship between samples and labels. In our framework, although we employ adversarial mask module to disentangle the spurious correlated non-causal information of domain-invariant features, it can not entirely eliminate the non-causal information due to the lack of diversity.

Therefore, we propose the causal related feature augmentation to create causal related hard features to enhance the diversity. To construct cross-class causal related hard features, we select a cross-class non-causal information, which is in the form of inferior dimensions

of domain-invariant features, for each disentangled feature. Intuitively, the spurious correlated non-causal information in one class may also exhibit a degree of spurious correlation with its similar categories. With this insight, the class with the greatest classification probability excluding its label class is selected as objective for non-causal information selection. Similar with the domain related feature augmentation, information entropy is served as the criterion for selecting non-causal information among objective classes. As illustrated in Eq 6, lower information entropy indicates a more certain spurious causal correlation. Specifically, to mitigate the impact of varying domain information, we implement feature augmentation within one domain. Within a specific domain, the information entropy criterion is leveraged to select a non-causal information feature for different categories, resulting in the selection of $C$ features for $C$ classification tasks. With $k$ source domains in total, the overall number of selected non-causal information features is $k \times C$. Subsequently, for each disentangled feature, we select the non-causal information feature corresponding to its target class from the $C$ non-causal features in its source domain. The chosen non-causal features and disentangled features are then merged via the concatenation operation and reduce the feature dimension through a fully connected layer following the above section operation, as shown in Eq 9. $f'_{inf}$ represents the inferior dimensions of domain-invariant features that belong to the objective selection class within a domain.

$$f_{CR-Aug} = \text{FC}([f_{sup}, min(IE(f'_{inf}))]) \quad (9)$$

The domain-invariant features contain the same causal information as the causal related hard features. To encourage the domain-invariant features to disregard the non-causal information, we employ supervised contrastive learning between domain-invariant features and causal related hard features, as shown in Eq 10.

$$\mathcal{L}_{cl}^{CR} = \ell_{cl}(f_I, f_{CR-Aug}) \quad (10)$$

The overall loss of the dual-stream feature augmentation is depicted in Eq 11.

$$\mathcal{L}_{cl} = \mathcal{L}_{cl}^{DR} + \mathcal{L}_{cl}^{CR} \quad (11)$$

## 3.4 Overall Training and Inference

The overall training process is composed of three components. The domain-specific encoder $F_s$ and domain classifier $C_d$ are updated according to Eq 12. The domain-invariant encoder $F_i$ and label classifier $C_1$, $C_2$ are updated according to Eq 13. The adversarial mask $M$ is updated according to Eq 14. $\lambda_{inv}$ and $\lambda_{cl}$ are the corresponding trade-off parameters.

$$\min_{\hat{F}_s, \hat{C}_d} \mathcal{L}_{dc}^{spe} \quad (12)$$

$$\min_{\hat{F}_i, \hat{C}_1, \hat{C}_2} \mathcal{L}_{cls} + \lambda_{inv}\mathcal{L}_{dc}^{inv} + \lambda_{cl}\mathcal{L}_{cl} \quad (13)$$

$$\min_{\hat{M}} \mathcal{L}_{mask} \quad (14)$$

During the inference, the parameters in model are fixed. Domain-invariant encoder $F_1$ and the label classifier $C_1$ are leveraged for inference.

### Table 1: leave-one-domain-out results on PACS

| Target | Art | Cartoon | Photo | Sketch | Ave. |
|---|---|---|---|---|---|
| ResNet18 | | | | | |
| DeelAll [54] | 77.63 | 76.77 | 95.85 | 69.50 | 79.94 |
| EISNet [43] | 81.89 | 76.44 | 95.93 | 74.33 | 82.15 |
| MixStyle [56] | 84.10 | 78.80 | 96.10 | 75.90 | 83.73 |
| MSAM [26] | 85.50 | 78.75 | **96.53** | 75.28 | 84.02 |
| FACT [46] | 85.37 | 78.38 | 95.15 | 79.15 | 84.51 |
| MatchDG [29] | 81.32 | 80.70 | **96.53** | 79.72 | 84.56 |
| DSON [35] | 84.67 | 77.65 | 95.87 | 82.83 | 85.11 |
| RSC [16] | 83.43 | 80.31 | 95.99 | 80.85 | 85.15 |
| IPCL [8] | 85.35 | 78.88 | 95.63 | 81.75 | 85.40 |
| StyleNeo [18] | 84.41 | 79.25 | 94.93 | **83.27** | 85.47 |
| FSDCL [17] | 85.30 | 81.31 | 95.63 | 81.19 | 85.86 |
| FSR [44] | 84.49 | 81.15 | 96.13 | 82.01 | 85.95 |
| FFDI [41] | 85.2 | 81.5 | 95.8 | 82.8 | 86.3 |
| CIRL [28] | 86.08 | 80.59 | 95.93 | 82.67 | 86.32 |
| XDED [21] | 85.60 | **84.20** | 96.50 | 79.10 | 86.40 |
| DFA(ours) | **87.20** | 80.88 | 96.22 | 82.92 | **86.80** |
| ResNet50 | | | | | |
| mDSDI [3] | 87.70 | 80.40 | **98.10** | 78.40 | 86.20 |
| FSDCL [17] | 88.48 | 83.83 | 96.59 | 82.92 | 87.96 |
| CCFP [22] | - | - | - | - | 88.40 |
| PCL [47] | 90.20 | 83.90 | **98.10** | 82.60 | 88.70 |
| FFDI [41] | 89.30 | 84.70 | 97.10 | 83.90 | 88.80 |
| StyleNeo [18] | 90.35 | 84.2 | 96.73 | 85.18 | 89.11 |
| FACT [46] | **90.89** | 83.65 | 97.78 | 86.17 | 89.62 |
| CIRL [28] | 90.67 | 84.30 | 97.84 | **87.68** | 90.12 |
| DFA(ours) | 90.62 | **85.87** | 97.60 | 87.52 | **90.40** |

## 4 EXPERIMENTS

### 4.1 Dataset

To verify the effectiveness of the proposed method, we evaluate our method on four public datasets, which cover various recognition scenes. **PACS** [24] is a public object recognition dataset which has large discrepancy in different domains. It contains 999,1 images from four domains (Art-Painting, Cartoon, Photo and Sketch), and in each domain, it contains 7 categories. For fair comparison, we follow the original training-validation split provided by [24]. **OfficeHome** [40] is a large public dataset with 4 domains, and each domain consists of 65 categories. The four domains are Art, Clipart, Product and Real-World. It contains 15,500 images, with an average of around 70 images per class. Following [28], we randomly split each domain into 90% for training and 10% for validation. **VLCS** [14] is a mixture of different datasets, named as VOC2007 [11], LabelMe [34], Caltech101 [12] and SUN09 [9]. Each domain contains 5 categories. Following [8], we randomly split 80% for training and 20% for validation. **TerraIncognita** [2] is a very large dataset includeing 24,778 photographs of wild animals, which are divided into 10 categories. It contains 4 camera-trap domains: L100, L38, L43, L46.

**Table 2: leave-one-domain-out results on OfficeHome**

| Target | Art | Clipart | Product | Real | Avg. |
|---|---|---|---|---|---|
| | | ResNet18 | | | |
| DeepAll [54] | 57.88 | 52.72 | 73.50 | 74.80 | 64.72 |
| JiGen [5] | 53.04 | 47.51 | 71.47 | 72.79 | 61.20 |
| RSC [16] | 58.42 | 47.90 | 71.63 | 74.54 | 63.12 |
| MixStyle [56] | 57.20 | 52.90 | 73.50 | 75.30 | 64.87 |
| MSAM [26] | 57.56 | 53.54 | 73.76 | 75.97 | 65.21 |
| StyleNeo [18] | 59.55 | 55.01 | 73.57 | 75.52 | 65.89 |
| FSDCL [17] | 60.24 | 53.54 | 74.36 | 76.66 | 66.20 |
| IPCL [8] | 61.56 | 53.13 | 74.32 | 76.22 | 66.31 |
| FFDI [41] | **61.70** | 53.80 | 74.40 | 76.20 | 66.50 |
| FSR [44] | 59.95 | 55.07 | 74.82 | 76.34 | 66.55 |
| FACT [46] | 60.34 | 54.85 | 74.48 | 76.55 | 66.56 |
| CIRL [28] | 61.48 | 55.28 | 75.06 | 76.64 | 67.12 |
| DFA(ours) | 61.22 | **55.41** | **75.12** | **76.81** | **67.14** |
| | | ResNet50 | | | |
| MixStyle [56] | 51.1 | 53.2 | 68.2 | 69.2 | 60.4 |
| MLDG [23] | 61.5 | 53.2 | 75.0 | 77.5 | 66.8 |
| ERM [38] | 63.1 | 51.9 | 77.2 | 78.1 | 67.6 |
| SagNet [32] | 63.4 | 54.8 | 75.8 | 78.3 | 68.1 |
| CORAL [36] | 65.3 | 54.4 | 76.5 | 78.4 | 68.7 |
| mDSDI [3] | **68.1** | 52.1 | 76.0 | 80.4 | 69.2 |
| CCFP [22] | - | - | - | - | 69.7 |
| SWAD [6] | 66.1 | 57.7 | 78.4 | 80.2 | 70.6 |
| PCL [47] | 67.3 | 59.9 | 78.7 | **80.7** | 71.6 |
| DFA(ours) | 67.6 | **60.7** | **79.4** | 80.7 | **72.1** |

## 4.2 Implementation Details

ImageNet pretrained on ResNet [15] is used as our backbone. We train the network with SGD, batch size of 16 and weight decay of 5e-4 for 50 epochs. The initial learning rate is 0.001 and decayed by 0.1 at 80% of the total epochs. For all datasets, images are resized to $224 \times 224$. The standard augmentation protocol in [4] is followed, which consists of random resized cropping, horizontal flipping and color jittering. We also adopt the Fourier data augmentation as in [46] and construct different domain-specific encoder for different source domain. Following the commonly used leave-one-domain-out protocol [25], we specify one domain as the unseen target domain for evaluation and train with the remaining domains. The parameter $\lambda_{inv}$ of the domain classifier loss is set to 1 and use a sigmoid ramp-up strategy [31] with a length of 5 epochs following [28]. To promote greater stability during training, we apply identity operations to the mask throughout the initial five epochs as [28]. The parameter $\lambda_{cl}$ of the contrastive learning loss is set to 0.001 after the initial five epochs. Inspired by [19], the temperature parameter $\tau$ of $\ell_{cl}$ is set to 0.07.

## 4.3 Results

**Results on PACS** is shown in Table 1. We compare our method with previous SOTA methods. Our method surpasses CIRL [28] by 0.48% on ResNet18 and 0.28% on ResNet50, respectively. This

**Table 3: leave-one-domain-out results on VLCS with ResNet18**

| Target | V | L | C | S | Avg. |
|---|---|---|---|---|---|
| DeelAll [15] | 67.48 | 61.81 | 91.86 | 68.77 | 72.48 |
| JiGen [5] | 70.93 | 62.06 | 96.17 | 71.40 | 75.14 |
| FACT [46] | 71.83 | 64.38 | 92.79 | 73.28 | 75.57 |
| FSR [44] | 71.94 | 61.03 | 97.95 | 71.42 | 75.59 |
| RSC [16] | 73.81 | 62.51 | 96.21 | 72.10 | 76.16 |
| MMLD [30] | 73.01 | 62.20 | 97.01 | 72.49 | 76.18 |
| MSAM [26] | 76.31 | 63.74 | 97.64 | 69.34 | 76.76 |
| IPCL [8] | 74.47 | 66.83 | 92.51 | 73.25 | 76.77 |
| MVDG [50] | 75.26 | 63.79 | 98.40 | 71.05 | 77.13 |
| StableNet [51] | 73.59 | 65.36 | 96.67 | 74.97 | 77.65 |
| CIRL [28] | 73.04 | **68.22** | 92.93 | **77.27** | 77.87 |
| DFA(ours) | **76.45** | 67.00 | 97.38 | 72.51 | **78.33** |

**Table 4: leave-one-domain-out results on TerraIncognita with ResNet50**

| Target | L100 | L38 | L43 | L46 | Avg. |
|---|---|---|---|---|---|
| Mixstyle [56] | 54.3 | 34.1 | 55.9 | 31.7 | 44.0 |
| RSC [16] | 50.2 | 39.2 | 56.3 | 40.8 | 46.6 |
| DANN [13] | 51.1 | 40.6 | 57.4 | 37.7 | 46.7 |
| IRM [1] | 54.6 | 39.8 | 56.2 | 39.6 | 47.6 |
| CORAL [36] | 51.6 | 42.2 | 57.0 | 39.8 | 47.7 |
| MLDG [23] | 54.2 | 44.3 | 55.6 | 36.9 | 47.8 |
| ERM [38] | 54.3 | 42.5 | 55.6 | 38.8 | 47.8 |
| mDSDI [3] | 53.2 | 43.3 | 56.7 | 39.2 | 48.1 |
| SagNet [32] | 53.0 | 43.0 | 57.9 | 40.4 | 48.6 |
| SWAD [6] | 55.4 | 44.9 | 59.7 | 39.9 | 50.0 |
| PCL [47] | 58.7 | 46.3 | **60.0** | **43.6** | 52.1 |
| DFA(ours) | **59.9** | **50.2** | 57.0 | 42.8 | **52.5** |

improvement is attributed to learning causal information from domain-invariant information, thereby excluding causal-related but domain-specific information. Furthermore, our method outperforms data augmentation methods such as FSR [44] and FFDI [41], because we encompasses not only feature stylization augmentation but also causal-related feature augmentation. Specifically, compared with CCFP [22], which also adopt feature augmentation, DFA surpasses CCFP by 2%. Through dual-stream feature augmentation, both the transferability and discriminability of features are enhanced. Our method achieves the best performance, achieving an average accuracy of 86.80% on ResNet18 and 90.40% on ResNet50. **Results on OfficeHome** is shown in Table 2 which illustrates that DFA outperforms data augmentation methods like FACT [46], FSR [44] and FFDI [41]. However, the impact of DFA on ResNet18 is limited due to the image number per category is small in this dataset and the data style is similar to its pretrained dataset ImageNet with a small domain gap. In such scenarios, some domain style information may enhance the classification results. On ResNet50, DFA is 2.9% higher than mDSDI [3] method, and 0.5% higher than PCL [47] method. **Results on VLCS** is shown in Table 3. we use ResNet18

**Table 5: An ablation study of baseline method and our DFA.**

| Model | DFD | AdvM | DR | CR | A | C | P | S | Avg. |
|-------|-----|------|----|----|------|------|------|------|------|
| Model1 | ✓ | - | - | - | 85.25 | 78.62 | 96.22 | 79.15 | 84.81 |
| Model2 | - | ✓ | - | - | 85.93 | 80.07 | 96.28 | 81.52 | 85.95 |
| Model3 | ✓ | ✓ | - | - | 86.57 | 79.94 | 96.28 | 81.92 | 86.17 |
| Model4 | ✓ | ✓ | ✓ | - | 87.06 | 80.46 | 96.46 | 82.28 | 86.56 |
| Model5 | ✓ | ✓ | - | ✓ | 86.86 | 80.33 | 96.22 | 82.64 | 86.51 |
| DFA | ✓ | ✓ | ✓ | ✓ | 87.20 | 80.88 | 96.22 | 82.92 | 86.80 |

as the backbone and our DFA demonstrates superior performance, outperforming CIRL [28] by an average of 0.46%, and surpassing FSR [44] by an average of 2.74%. **Results on TerraIncognita** is shown in Table 4 and ResNet50 is employed as the backbone. Our DFA exhibits superior performance, surpassing mDSDI [3] by an average margin of 4.4%. DFA outperforms PCL [47] by an average of 0.4% which is a robust contrastive learning method.

Based on the above results from four benchmarks in DG task, DFA outperforms other data augmentation methods, particularly in situations with large domain gaps. DFA achieves the feature augmentation by considering both domain-specific information and causally correlated information, thereby the generalization capability of the model is improved.

## 5 DISCUSSION

**Ablation Study.** We conduct ablation studies to demonstrate the significance of each module in Table 5. "DFD" and "AdvM" represent dual path feature disentangle module and adversarial mask module, respectively. "DR" and "CR" represent domain-related feature augmentation and causal-related feature augmentation, respectively. We employ ResNet18 as the backbone and train on the PACS dataset. Firstly, we discuss the ablation study of the baseline ( corresponds to model3 in the table) which represents the feature disentanglement framework without the hard feature perturbation. Comparing model3 with model1 and model2, it is obvious that the performance of combining both Adversarial Training and Adversarial Mask is much better. This observation suggests that for DG problems, it is insufficient to learn only domain-invariant features or causal features. Rather, considering causal information within domain-invariant features can directly improve model performance. Additionally, the performance enhancements seen in Model4 and Model5 indicate that the two types of feature augmentation methods we proposed can help the model concentrate on hard features, thereby improving the model's ability to discriminate hard features. Finally, based on the baseline, the DFA achieves the SOTA result of 86.80%, demonstrating that the two types of feature augmentation methods further enhance the transferability and discriminability of features.

**Analysis with GradCAM.** In Figure 4, we visualize the attention maps of the last convolutional layer to verify the efficacy of causal related feature augmentation. The second row presents the baseline, while the last row demonstrating the efficacy of causal related feature augmentation. It is evident that, despite the baseline achieving relatively satisfactory test results, it still encounters challenges with samples that have spurious correlations. It fails to capture the causal information, instead focusing on non-causal

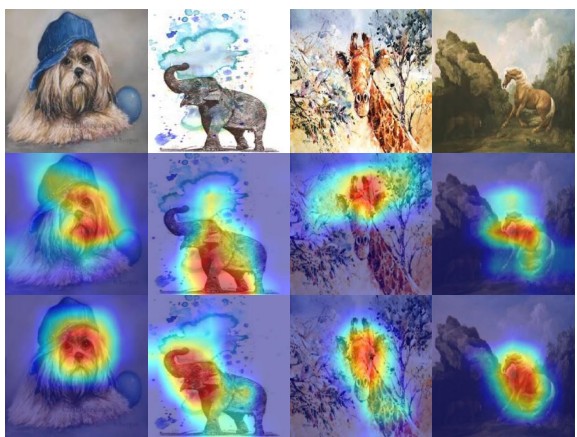

**Figure 4: Visualization of attention maps of the last convolutional layer for our base framework and DFA. We use ResNet18 as the backbone and train on the PACS dataset, with Art Painting serving as the target domain.**

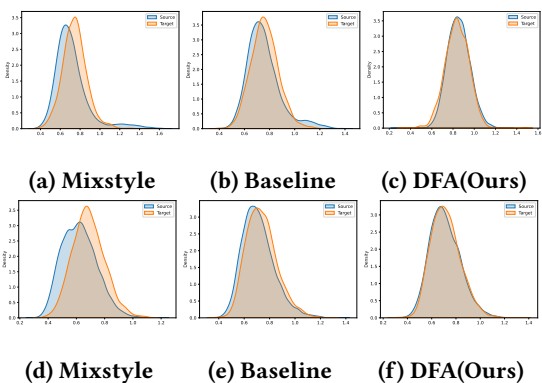

| (a) Mixstyle | (b) Baseline | (c) DFA(Ours) |
|---|---|---|
| (d) Mixstyle | (e) Baseline | (f) DFA(Ours) |

**Figure 5: The visualization of feature statistics. The top raw is the mean statistics and the bottom raw is the std statistics. We use ResNet18 as the backbone and train on the PACS dataset, with Art Painting serving as the target domain.**

information, as demonstrated in Figure 4. These results suggest that causal related feature augmentation can effectively enhance the model's ability to identify the causal information of samples, consequently strengthening discriminability of features.

**Analysis of Feature Statistics.** To confirm that domain related feature augmentation can effectively mitigate the effects of domain shift, we visualize the feature statistics distribution based on Mixstyle, baseline and DFA. Compared with Mixstyle [56], the baseline successfully learns domain-invariant information, exhibiting minimal shifts in feature statistics and our feature augmentation clearly mitigates the domain shift between different domain features, indicating a higher purity of domain-invariant features.

**Confusion Matrix.** We have plotted confusion matrix for our baseline and DFA, as illustrated in Fig. 7. We employ ResNet18 as backbone and train on the PACS dataset.It can be obviously found that in the art and cartoon domains, the baseline still has

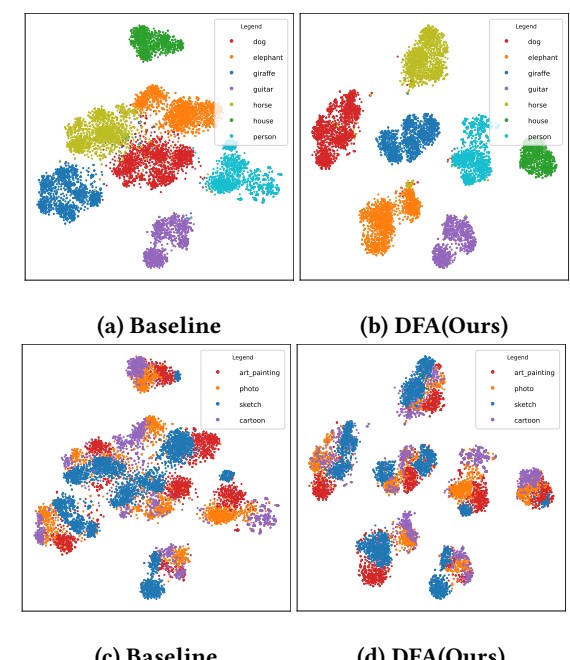

(a) Baseline                (b) DFA(Ours)

(c) Baseline                (d) DFA(Ours)

**Figure 6: The t-SNE visualization of feature representations extracted by the feature extractor of the baseline and DFA on PACS. Different colors mean different classes in (a) and (b), and different domains in (c) and (d), respectively.**

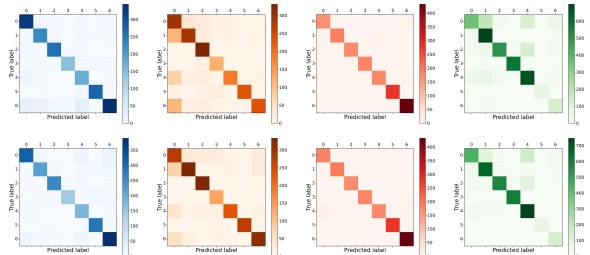

**Figure 7: The confusion matrix of baseline and DFA. Each color represents a target domain, ordered from left to right as follows: Art, Cartoon, Photo, Sketch. The top row is the baseline, and the bottom row is our DFA.**

some incorrect classifications due to non-causal information and domain shift. In contrast, DFA displays a significant reduction in classification errors. This evidence suggests that DFA is capable of eliminating such spurious correlations in samples and paying attention to domain-invariant and causally related information, thereby enhancing the model's generalization ability.

**Visualization of Features.** We employ t-SNE [37] to display the visualization results of features extracted by the semantic feature extractor, as depicted in Fig. 6. From Fig. 6(a), where different colors denote different classes, it becomes clear that although the baseline can distinguish each category in the feature space, it still struggles to differentiate samples with similar semantics. This challenge is indicated by a mixture of points from different class labels

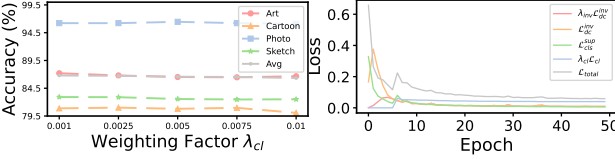

(a) Parameter sensitivity        (b) Loss decline curve

**Figure 8: (a) is the sensitivity analysis of the parameter $\lambda_{cl}$. (b) is loss curve of training process. All results are obtained on PACS dataset with ResNet18 as backbone.**

in the middle. Considering that horses, elephants, and dogs are all tetrapods, the baseline fails to capture information that determines class labels, such as elephants' trunk, ears, and instead focuses on non-causal information such as body or legs. Through DFA, we can construct more hard features to train the model, thereby eliminating the spurious correlations contained in semantic features, as evidenced in Fig. 6(b). Compared with Fig. 6(c) and 6(d), DFA can reduce the distance between different domains in the feature space, enabling the model to learn domain-invariant features and eliminate potential domain-specific information. Thus, these results reveal that DFA is indeed capable of directing the feature extractor to focus more on domain-invariant and causal related information, thereby enhancing the model's generalization capability on unseen target domains.

**Parameter Sensitivity.** We analyze the sensitivity of Parameter $\lambda_{cl}$ on the PACS dataset with ResNet18 as the backbone, as depicted in Fig. 8(a). DFA robustly achieves competitive performances across a broad range of values. Fig. 8(b) illustrates the loss decline curve, where $\lambda_{cl}$ is 0.005 and $\lambda_{inv}$ employs a sigmoid ramp-up [31] with a length of 5 epochs. The orange line in the Fig. 8(b) is converging quickly which is the domain classification loss of domain-invariant features. It indicates that our dual path disentangle module can learn domain-invariant features in a significantly more stable manner, demonstrating an advantage compared to traditional domain adversarial training. The entire training process exhibits stability, with both $\mathcal{L}_{dc}^{inv}$ and $\mathcal{L}_{cl}$ converging, indicating that our method provides a stable end-to-end framework.

# 6 CONCLUSION

In this paper, a dual stream feature augmentation is proposed based on the disentanglement framework. Previous work always applied random perturbations on style, however, they do not exploit potentialities of feature transferability. Differently, on the one hand, we construct domain related hard features to explore harder and broader style spaces while preserving semantic consistency. On the other hand, considering that the spurious correlated non-causal information can harm the discriminability of model, the causal related hard features are also constructed to better disentangle the non-causal information hidden in domain-invariant features, thereby improving the generalization and robustness of the model. Trough dual-stream feature augmentation based on a stable feature disentanglement framework, we successfully learn causal related domain-invariant features, and a variety of experiments demonstrate the effectiveness of our method. In the future, we will try to integrate our work with the challenging multimodel learning task.

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
