# OpenReview forum: "Dual-stream Feature Augmentation for Domain Generalization"
_acmmm.org/ACMMM/2024/Conference — MM2024 Poster_

### Official Review · Reviewer_5T1x · 2024-05-23

**Rating:** 4
**Confidence:** 3

**Summary:**

To address the current issues in domain generalization, such as insufficient exploration of extreme perturbed features and the neglect of causal relationships in discriminative features, the paper proposes a Dual-stream Feature Augmentation framework. The framework improves the model's generalization ability to unseen domains by integrating domain-relevant and causality-aware hard feature generation.

**Strengths:**

- Proposing a novel domain-relevant hard feature perturbation strategy. The strategy is based on an end-to-end stable disentanglement framework, which not only maintains the semantic consistency of the data but also effectively enhances feature transferability. This approach offers new insights for improving model generalization in unseen domains.

- Exploring and enhancing the distinguishability of generalized features by introducing causality. By creating causal-related hard features through the disentanglement framework, it eliminates the underlying non-causal information hidden in the features, thereby improving adaptability to complex data environments.

- Extensive experiments conduct on multiple public benchmarks clearly demonstrate the effectiveness of the proposed method.

**Limitations:**

- In section 3.3.1, the authors state "Lower information entropy indicates a more distinct style information" and in section 3.3.2, it states "Lower information entropy indicates a more certain spurious causal correlations". Both sections use information entropy to measure style differences and causal spuriousness across different domains. Could the authors provide a more explicit theoretical basis and explanation for the direct association between low information entropy and the uniqueness of style or spuriousness of causal relationships?

- What baseline or threshold is "Lower" in "Lower information entropy" relative to? If there is a baseline or threshold, are the standards for using information entropy consistent across both instances? Could authors provide a detailed explanation.

- Although the authors conduct ablation experiments to validate the innovations proposed in the paper from a comprehensive performance perspective, the related analysis is lacking. It is recommended that the authors enhance the analysis and discussion in the experimental section.

- References [24] and [25] are the same document. Please check for citation errors in the references.

- It is suggested to add more details to Figure 3, such as "CL"

**Suitability:**

2

---

### Official Review · Reviewer_XmvA · 2024-05-23

**Rating:** 3
**Confidence:** 2

**Summary:**

This paper proposes two augmentations from the perspectives of transferability and causality respectively and integrated them into a unified learnable model to solve existing limitations of DG methods. Experiments on classical DG datasets its effectiveness.

**Strengths:**

1.	The method achieves SOTA on several classical datasets as reported in the paper.
2.	The algorithm design makes sense, and the specific implementation demonstrates thoughtful design elements that are interesting.
3.	Figure 1 is clear.

**Limitations:**

1. The author is attempting to explain their method in detail. However, some expression in "Method" section is still rather unclear. For example, why “the semantic information of domain related hard features aligns with domain-invariant features”?
2. The experiments are not comprehensive enough. For example, VLCS is only conducted with ResNet18 while TerraIncognita only have ResNet50 results, which is not a mainstream implementation in DG.
3. Some demonstrations are not clear enough. The caption of Figure 2 is confusing. Figure 4 may need to include more attention maps about essential baselines other than ERM. The analysis regarding the confusion matrix and Figure 7 in discussion seems to offer limited value.

**Suitability:**

3

---

### Official Review · Reviewer_Up4s · 2024-05-23

**Rating:** 4
**Confidence:** 3

**Summary:**

Traditional DG methods often fall short because they do not adequately explore feature transferability due to the randomness of feature augmentation, and they fail to account for causality in discriminative features, which leads to issues with spurious correlations.

The DFA method addresses these problems through two main strategies. First, it improves feature transferability by creating targeted features with domain-related augmentations, guided by uncertainty to simulate hard cross-domain scenarios. Second, it incorporates causality by using an adversarial mask to disentangle spurious, non-causal information, allowing for the extraction of more discriminative, causal-related features.

This approach differs from previous methods by integrating the two types of augmentations into a unified, learnable model with a disentangled feature strategy. Additionally, contrastive learning is employed to maintain semantic consistency and enhance the model's robustness. Extensive experiments on several datasets have shown that the DFA method achieves state-of-the-art performance in domain generalization, demonstrating its effectiveness in overcoming traditional DG challenges.

**Strengths:**

The authors focus on the domain related hard features and make adversarial augmentations to boost the performance of DG.

**Limitations:**

The computational cost may be larger than other methods.

**Suitability:**

2

---

### Official Review · Reviewer_YhGH · 2024-05-24

**Rating:** 4
**Confidence:** 3

**Summary:**

The paper presents a method aimed at constructing hard features from two perspectives: domain-related and causal-related, to improve the model's transferability and discriminability. It integrates these augmentations into a unified learnable model with a disentangled feature strategy and employs contrastive learning to maintain semantic consistency and robustness.

**Strengths:**

1. Introduces a novel dual-stream augmentation strategy for improving generalization across domains.

2.  Addresses the issue of spurious correlations by disentangling causal and non-causal information.

3. Provides extensive experimental results on various datasets to validate the effectiveness of the proposed method.

**Limitations:**

1. The experimental section is somewhat incomplete, with the following issues:
   - The baselines for comparison are not explained.
   - The comparative work lacks the most recent research efforts, with most being from 2022 and earlier.
   - There are no results for VLCS using the ResNet50 backbone.

2. The choice of backbone: Why are methods still using ResNet18 as the backbone for comparison? For recent DG tasks, isn't the default backbone usually ResNet50 or Vision Transformer?

3. Why is the weight of $\lambda_\text{cl}$ so low? Does $l_\text{cl}$ really play a role?

**Suitability:**

3

---

### Meta-Review · Area_Chair_sabD · 2024-07-01

**Recommendation:** Accept (Poster)
**Confidence:** 4

**Metareview:**

The reviews recognize the paper's innovative Dual-stream Feature Augmentation (DFA) framework, which enhances domain generalization by integrating domain-relevant and causality-aware feature generation. This approach effectively addresses spurious correlations and improves feature transferability, as demonstrated by extensive experiments across various datasets. While some reviewers noted minor issues with experimental details and methodological clarity, these concerns are outweighed by the paper's significant contributions and novel strategies. Therefore, despite the need for minor refinements, the paper merits acceptance.